# Bootstrapping Neural Processes

**Juho Lee**[1,2*], **Yoonho Lee**[2*], **Jungtaek Kim**[3],
**Eunho Yang**[1,2], **Sung Ju Hwang**[1,2], **Yee Whye Teh**[4]
KAIST[1], Daejeon, South Korea, AITRICS[2], Seoul, South Korea,
POSTECH[3], Pohang, South Korea, University of Oxford[4], Oxford, England
juholee@kaist.ac.kr

## Abstract

Unlike in the traditional statistical modeling for which a user typically hand-specify a prior, Neural Processs (NPs) implicitly define a broad class of stochastic processes with neural networks. Given a data stream, NP learns a stochastic process that best describes the data. While this "data-driven" way of learning stochastic processes has proven to handle various types of data, NPs still rely on an assumption that uncertainty in stochastic processes is modeled by a single latent variable, which potentially limits the flexibility. To this end, we propose the Bootstrapping Neural Process (BNP), a novel extension of the NP family using the bootstrap. The bootstrap is a classical data-driven technique for estimating uncertainty, which allows BNP to learn the stochasticity in NPs without assuming a particular form. We demonstrate the efficacy of BNP on various types of data and its robustness in the presence of model-data mismatch.

## 1 Introduction

Neural Process (NP) [8] is a class of stochastic processes defined by parametric neural networks. Traditional stochastic processes such as Gaussian Process (GP) [19] are usually derived from mathematical objects based on certain prior beliefs on data (e.g., smoothness of functions quantified by Gaussian distributions). On the other hand, given a stream of data, NP *learns* to construct a stochastic process that might describe the data well. In that sense, NP may be considered as a data-driven way of defining stochastic processes. When appropriately trained, NP can define a flexible class of stochastic processes well suited for highly non-trivial functions that are not easily represented by existing stochastic processes.

Like other stochastic processes, NP induces stochasticity in function realizations. More specifically, NP defines a function value $y$ for a point $x$ as a conditional distribution $p(y|x, \dots)$ to model *pointwise uncertainty*. Additionally, NP further introduces a *global latent variable* capturing *functional uncertainty* - a global uncertainty in the overal structure of the function. The global latent variable modeling functional uncertainty is empirically demonstrated to improve the predictive performance and diversity in function realizations [14].

Although it is clear both intuitively and empirically that adding functional uncertainty helps, it remains unclear whether modeling it with a single Gaussian latent variable is optimal. For instance, [16] pointed out that the global latent variable acts as a bottleneck. One could introduce more complex architectures to better capture the functional uncertainty, but that would typically come with an architectural overhead. Moreover, it contradicts the philosophy behind NP to use minimal modeling assumptions and let the model learn from data.

---

This paper introduces a novel way of introducing functional uncertainty to the family of NP models. We revisit the bootstrap [6], a classic frequentist technique to model uncertainty in parameter estimation by simulating population distribution via resampling. The bootstrap is a simple yet effective way of modelling uncertainty in a data-driven way, making it well-suited for our purpose of giving uncertainty to NP with minimal modeling assumptions. To this end, we propose BNP, an extension of NP using bootstrap to induce functional uncertainty. BNP utilizes bootstrap to construct multiple resampled datasets and combines the predictions computed from them. The functional uncertainty is then naturally induced by the uncertainty in the bootstrap procedure.

BNP can be defined for any existing NP variants with minimal additional parameters and provides several benefits over existing models. One important aspect is its robustness under the presence of *model-data mismatch*, where test data come from distributions different from the one used to train the model. An ensemble of bootstrap is well known to enhance the stability and accuracy [1]. Recently, [11] showed that ensembling Bayesian posteriors from multiple bootstrap samples dramatically improves the robustness under model-data mismatch. We show that our extension of NP with bootstrap also enjoys this property. Using various data ranging from simple synthetic data to challenging real-world data, we demonstrate that BNP is much more robust than the existing NP with global latent variables. This tendency was particularly strong under model-data mismatch, where the test data is significantly different from the datasets used to train the model.

## 2 Background

### 2.1 (Attentive) Neural Processes

Consider a regression task $\mathcal{T} = (X, Y, c)$ defined by an observation set $X = \{x_i\}_{i=1}^n$, a label set $Y = \{y_i\}_{i=1}^n$, and an index set $c \subsetneq \{1, \ldots, n\}$ defining *context* $(X_c, Y_c) := \{(x_i, y_i)\}_{i \in c}$. The goal is to learn a stochastic process (random function) mapping $x$ to $y$ given the context $(X_c, Y_c)$ as training data (a realization from the stochastic process), i.e., learning

$$\log p(Y|X, Y_c) = \sum_{i=1}^n \log p(y_i|x_i, X_c, Y_c). \tag{1}$$

Conditional Neural Process (CNP) [7] models $p(y_i|x_i, X_c, Y_c)$ with a deterministic neural network taking $(X_c, Y_c)$ and $x_i$ to output the parameters of $p(y_i|x_i, X_c, Y_c)$. CNP consists of an encoder and a decoder; the encoder summarizes $(X_c, Y_c)$ into a representation $\phi$ via permutation-invariant neural network [5, 25], and the decoder transforms $\phi$ and $x_i$ into the target distribution (e.g., Gaussian),

$$\phi = f_{\text{enc}}(X_c, Y_c) = f_{\text{enc}}^{(2)}\left(\frac{1}{|c|} \sum_{i \in c} f_{\text{enc}}^{(1)}(x_i, y_i)\right), \tag{2}$$

$$(\mu_i, \sigma_i) = f_{\text{dec}}(\phi, x_i), \quad p(y_i|x_i, X_c, Y_c) = \mathcal{N}(y_i|\mu_i, \sigma_i^2), \tag{3}$$

where $f_{\text{enc}}^{(1)}, f_{\text{enc}}^{(2)}$ and $f_{\text{dec}}$ are feed-forward neural networks. CNP is then trained to maximize the expected likelihood $\mathbb{E}_{p(\mathcal{T})}[\log p(Y|X, Y_c)]$. The variance $\sigma_i^2$ models the *point-wise* uncertainty for $y_i$ given the context. NP [8] further models *functional uncertainty* using a *global latent variable*. Unlike CNP, which maps a context into a deterministic representation $\phi$, NP encodes the context into a Gaussian latent variable $z$, giving additional stochasticity in function construction. Following [12], we consider a NP with both deterministic path and latent path, where the deterministic path models the overall skeleton of the function $\phi$, and the latent path models the functional uncertainty:

$$\phi = f_{\text{denc}}(X_c, Y_c), \quad (\eta, \rho) = f_{\text{lenc}}(X_c, Y_c), \quad q(z|X_c, Y_c) = \mathcal{N}(z; \eta, \rho^2) \tag{4}$$

$$(\mu_i, \sigma_i) = f_{\text{dec}}(\phi, z, x_i), \quad p(y_i|x_i, z, \phi) = \mathcal{N}(y_i|\mu_i, \sigma_i^2), \tag{5}$$

with $f_{\text{denc}}$ and $f_{\text{lenc}}$ having the same structure as $f_{\text{enc}}$ in (2). The conditional probability is lower-bounded as

$$\log p(Y|X, Y_c) \geq \sum_{i=1}^n \mathbb{E}_{q(z|X,Y)}\left[\log \frac{p(y_i|x_i, z, \phi)p(z|X_c, Y_c)}{q(z|X, Y)}\right]. \tag{6}$$

We further approximate $p(z|X_c, Y_c) \approx q(z|X_c, Y_c)$ and train the model by maximizing this expected lower-bound over tasks.

Attentive Neural Process (ANP) [12] and its conditional version without a global latent variable, Conditional Attentive Neural Process (CANP), both employ an attention mechanism [22] to resolve the issue of underfitting in the vanila NP model. The encoder in ANP utilizes self-attention and cross-attention operation to better summarize the context into a representation $\phi$. Please refer to Appendix A for a detailed description about the architectures.

## 2.2 Bootstrap, Bagging, and BayesBag

Let $X = \{x_i\}_{i=1}^n$ be a dataset and $\theta = F(X)$ a parameter to estimate. Bootstrap [6] is a method that estimates the sampling distribution of $\theta$ from multiple datasets resampled from $X$,

$$\tilde{X}^{(j)} \overset{\text{s.w.r.}}{\sim} X, \quad \tilde{\theta}^{(j)} = F(\tilde{X}^{(j)}) \text{ for } j = 1, \dots, k, \tag{7}$$

where $\overset{\text{s.w.r.}}{\sim}$ denotes sampling with replacement [1]. We call each $\tilde{X}^{(j)}$ a *bootstrap dataset* and $\tilde{\theta}^{(j)}$ a *bootstrap estimate*. The bootstrap estimates are used for assessing uncertainty, computing credible intervals, or statistical testing. One can interpret the bootstrap estimates as samples from an (approximate) nonparametric and noninformative posterior of $\theta$ [10, page 272]. Contrary to standard Bayesian methods that specify an explicit prior $p(\theta)$, bootstrapping is a more "data-driven" way of computing the uncertainty of $\theta$.

**B**ootstrap **agg**regat**ing** (bagging) [1] is a procedure that ensembles multiple predictors given by bootstrap estimates. Let $T(\theta)$ be a predictor based on a parameter $\theta$, and $\{\tilde{\theta}^{(j)}\}_{j=1}^k$ be bootstrap estimates. The bagging predictor is computed as $\frac{1}{k}\sum_{j=1}^k T(\tilde{\theta}^{(j)})$. Bagging is known to improve accuracy and stability on classification and regression problems [1].

Instead of point estimates $T(\theta)$, one can also apply bagging to *Bayesian posteriors* $p(T(\theta)|X)$. BayesBag [4, 11] ensembles posteriors $\{p(T(\theta)|\tilde{X}^{(j)})\}_{j=1}^k$ computed from bootstrapped datasets to get an aggregated posterior $\frac{1}{k}\sum_{j=1}^k p(T(\theta)|\tilde{X}^{(k)})$. Compared to bagging, BayesBag provides similar or often better results even with fewer bootstrap datasets and is more robust under model-data mismatch [11].

## 2.3 Residual Bootstrap

Consider the bootstrap for regression, where a dataset is $(X, Y) = \{(x_i, y_i)\}_{i=1}^n$ and we want to estimate the distribution of the regression parameters $\theta$ or the predictive distribution $p(y|x, \theta)$. The most straightforward way is the paired bootstrap (empirical bootstrap) where we resample pairs of $(x, y)$ with replacement: $\{(\tilde{x}_i, \tilde{y}_i)\}_{i=1}^n \overset{\text{s.w.r.}}{\sim} \{(x_i, y_i)\}_{i=1}^n$. Unfortunately, since the probability of a pair $(x_i, y_i)$ being excluded in $(\tilde{X}, \tilde{Y})$ is approximately $(1 - n^{-1})^n \overset{n \to \infty}{\to} 0.368$, influential observations are often discarded, degrading the predictive accuracy.

Another option is the *residual bootstrap* which fixes $X$ and only resamples the residuals of predictions. Consider a nonparametric regression setting with prediction $\mu_i$, variance $\sigma_i^2$, and additive residual $\varepsilon_i$ ($\mu_i$ and $\sigma_i$ are functions of $x_i$), i.e., $y_i = \mu_i + \sigma_i\varepsilon_i$. Then, the bootstrap datasets are resampled as

1. Fit a model with $(X, Y)$ to obtain $\{(\mu_i, \sigma_i)\}_{i=1}^n$ and compute the residual $\varepsilon_i = \frac{y_i - \mu_i}{\sigma_i}$.

2. Let $\mathcal{E} = \{\varepsilon_i\}_{i=1}^n$, For $j = 1, \dots, k$,

   (a) Resample the residuals: $\tilde{\varepsilon}_1^{(j)}, \dots, \tilde{\varepsilon}_n^{(j)} \overset{\text{s.w.r.}}{\sim} \mathcal{E}$.

   (b) Construct a bootstrap dataset: for $i = 1, \dots, n$, $\tilde{x}_i^{(j)} = x_i$, $\tilde{y}_i^{(j)} = \mu_i + \sigma_i\tilde{\varepsilon}^{(j)}$.

The residual bootstrap resolves the issue of missing $x$ in bootstrap datasets, which is why they are often recommended for regression problems. We focus on using the residual bootstrap for our purpose, but one may also consider alternative bootstrap variants (e.g., wild bootstrap, parametric bootstrap) to resample datasets.

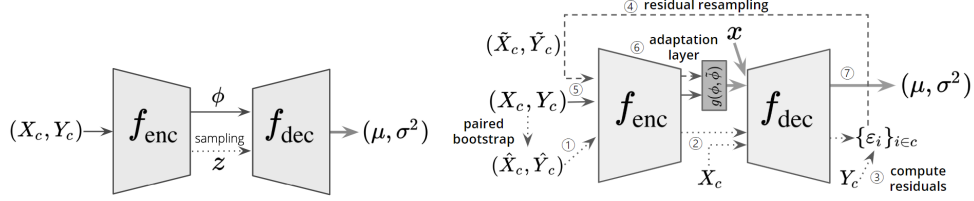

**Figure 1:** Diagrams for NP (left) and BNP (right).

# 3 Bootstrapping Neural Processes

## 3.1 Naïve application of residual bootstrap to NP does not work

One may consider directly applying residual bootstrap to existing NP models. That is, given a task $\mathcal{T} = (X, Y, c)$ and a NP model trained ordinarily, we can directly apply the residual bootstrap procedure described in Section 2.3 to get bootstrap contexts, and then compute bagged predictions by forwarding the bootstrap contexts through the NP model. NP is especially well-suited to this procedure because of its *amortization* in the inference step – it computes conditional probability $p(y|x, X_c, Y_c)$ efficiently as forward passes through neural networks. However, unfortunately, we found this works poorly in terms of predictive accuracy (Table D.5). This may be because 1) the amortization is suboptimal, making the errors from fitting multiple bootstrap datasets accumulate, and 2) the NP model does not see bootstrap datasets during training, so feeding bootstrap datasets through the network acts like a model-data mismatch scenario that can fool the model.

## 3.2 Bootstrapping Neural Processes

Beyond naïvely applying bootstrap to NP, we propose a novel class of NP called Bootstrapping Neural Process (BNP) which explicitly uses bootstrap datasets as additional inputs to induce functional uncertainty. BNP uses the NP as its "base" model, and the extension to ANP which we name Bootstrapping Attentive Neural Process (BANP) is defined similarly. Let $f_{\text{enc}}$ and $f_{\text{dec}}$ be encoder and decoder of a base NP (defined as in (2)), and $\mathcal{T} = (X, Y, c)$ be a task. BNP computes predictions through the following steps.

**Resampling contexts via paired bootstrap** Before proceeding to residual bootstrap, we first resample the contexts from $(X_c, Y_c)$ via paired bootstrap, that is, for $j = 1, \dots, k$,

$$(\hat{X}^{(j)}, \hat{Y}^{(j)}) := \{(\hat{x}_i^{(j)}, \hat{y}_i^{(j)})\}_{i=1}^{|c|} \overset{\text{s.w.r.}}{\sim} \{(x_i, y_i)\}_{i \in c}. \tag{8}$$

As noted in Section 2.3, some resampled context $(\hat{X}_c^{(j)}, \hat{Y}_c^{(j)})$ may miss several pairs from the original context. When passed to the model, such context would produce bad predictors, and thus large residuals. We empirically found that instead of computing single residuals computed from the full context $(X_c, Y_c)$ as in ordinary residual bootstrap, computing residuals from the multiple resampled contexts enhances robustness by exposing the model to residuals with diverse patterns during training. We present an ablation study comparing BNP with and without this step in Table D.5.

**Residual bootstrap** Now we do the inference for the full context $(X_c, Y_c)$ using the resampled contexts $(\hat{X}_c^{(j)}, \hat{Y}_c^{(j)})$. As noted above, this can be done efficiently by forwarding $(\hat{X}_c^{(j)}, \hat{Y}_c^{(j)})$ through $f_{\text{enc}}, f_{\text{dec}}$ to get $\{(\hat{\mu}_i, \hat{\sigma}_i)\}_{i \in c}$.

$$\hat{\phi}^{(j)} = f_{\text{enc}}(\hat{X}^{(j)}, \hat{Y}^{(j)}), \quad (\hat{\mu}_i^{(j)}, \hat{\sigma}_i^{(j)}) = f_{\text{dec}}(x_i, \hat{\phi}^{(j)}) \text{ for } i \in c. \tag{9}$$

Following the residual bootstrap procedure, we first compute residual, resample them,

$$\varepsilon_i^{(j)} = \frac{y_i - \hat{\mu}_i^{(j)}}{\hat{\sigma}_i^{(j)}} \text{ for } i \in c, \ \mathcal{E}^{(j)} = \{\varepsilon_i^{(j)}\}_{i=1}^{c}, \ \tilde{\varepsilon}_1^{(j)}, \dots, \tilde{\varepsilon}_{|c|}^{(j)} \overset{\text{s.w.r.}}{\sim} \mathcal{E}^{(j)}. \tag{10}$$

and construct bootstrap contexts to be used for the final prediction.

$$\tilde{x}_i^{(j)} = x_i, \ \tilde{y}_i^{(j)} = \hat{\mu}_i^{(j)} + \hat{\sigma}_i^{(j)} \tilde{\varepsilon}_i^{(j)} \text{ for } i \in c,$$
$$(\tilde{X}_c^{(j)}, \tilde{Y}_c^{(j)}) := \{(\tilde{x}_i^{(j)}, \tilde{y}_i^{(j)})\}_{i \in c} \text{ for } j = 1, \dots, k. \tag{11}$$

**Encoding with adaptation layer**   We pass the bootstrap contexts into the encoder to get the representations of the contexts, $\tilde{\phi}^{(j)} = f_{\text{enc}}(\tilde{X}_c^{(j)}, \tilde{Y}_c^{(j)})$ for $j = 1, \ldots, k$. The ordinary residual bootstrap would put each $\tilde{\phi}^{(j)}$ into the decoder and ensemble the decoded conditional probabilities. Instead, like NP using both deterministic representation $\phi$ and global latent variable $z$, we put both the representation of the original context $\phi = f_{\text{enc}}(X_c, Y_c)$ and the bootstrapped representation $\tilde{\phi}^{(j)}$ into the decoder. Since the decoder $f_{\text{dec}}$ is built to take only $\phi$, we add an *adaptation layer* $g(\phi, \tilde{\phi}^{(j)})$ to let $f_{\text{dec}}$ process a combined representation. The adaptation layer is the only part that we add to the base model, and can be implemented with a single linear layer. We empirically demonstrated that the adaptation layer is crucial for accurate prediction (Table D.5).

**Prediction**   Finally, we construct predictions by ensembling the predictions decoded from the representations of bootstrap contexts. For a target point $x_i$,

$$(\mu_i^{(j)}, \sigma_i^{(j)}) = f_{\text{dec}}(g(\phi, \tilde{\phi}^{(j)}), x_i), \quad p(y_i|x_i, \phi, \tilde{\phi}^{(j)}) = \mathcal{N}(y_i|\mu_i^{(j)}, (\sigma_i^{(j)})^2). \tag{12}$$

We compute this for $j = 1, \ldots, k$ to get an ensembled distribution,

$$p(y_i|x_i, X_c, Y_c) \approx \frac{1}{k}\sum_{j=1}^{k}\mathcal{N}(y_i|\mu_i^{(j)}, (\sigma_i^{(j)})^2). \tag{13}$$

Fig. 1 shows diagrams comparing NP and BNP. BNP uses almost the same architecture except for the adaptation layer, but goes through the encoding-decoding process twice (first to compute residuals only using the base model, and second to compute prediction with the adaptation layer added).

### 3.3   Training

BNP requires special care for training because we need to balance the training of the base model (without bootstrap) and the full model (with bootstrap). If we only train the full model, the decoder of the base model computing the residuals would produce inaccurate predictions yielding large residuals, making the full model likely to ignore the residual path during the early training stages. To resolve this, we train the model with a combined objective to simultaneously train two paths as follows,

$$\mathbb{E}_{p(\mathcal{T})}\left[\sum_{i=1}^{n}\left(\log p_{\text{base}}(y_i|x_i, X_c, Y_c) + \log\frac{1}{k}\sum_{j=1}^{k}\mathcal{N}(y_i|\mu_i^{(j)}, (\sigma_i^{(j)})^2)\right)\right], \tag{14}$$

where $p_{\text{base}}(y_i|x_i, X_c, Y_c)$ denotes the conditional probability computed from the base model (see Table D.5 for the ablation study). We also found that training with multiple bootstrap contexts (13) ($k > 1$) is crucial for robustness. We fixed $k = 4$ for all of our experiments.

### 3.4   Discussion

**Parallel computation**   An advantage of bootstrap and bagging is the ease in parallelization of fitting multiple bootstrap datasets. Our model also enjoys such benefits: we compute all steps (8)-(11) in parallel by packing multiple bootstrap contexts into a tensor and feeding it through networks.

**Our model and BayesBag**   Note that we are computing the aggregated conditional probability (13), which is similar to how BayesBag computes the aggregated posterior. The difference is that we aggregate the approximate distributions computed with a shared neural network ($f_{\text{enc}}$ and $f_{\text{dec}}$) while BayesBag independently computes posteriors. Although the theory in [11] does not directly apply to BNP, the underlying intuition may still be valid for our model: the predictions computed from BayesBag is more conservative (and thus robust) because it combines the model's uncertainty with the data-driven uncertainty coming from bootstrap.

**Why should NP be robust?**   Although we do not have theoretical claims that explain our model's robustness, we have intuitive explanations for such properties. When a BNP model encounters a substantial shift in data distribution, the base model will fail, resulting in larger residuals than usual. These larger residuals will be reflected in bootstrap contexts and thus into the representations $\tilde{\phi}^{(j)}$. This encourages the model to produce more conservative (larger $\sigma_i^2$) results (e.g, Fig. 2).

**Table 1:** 1D regression results. "context" refers to context log-likelihoods, and "target" refers to target log-likelihoods. Means and standard deviations of five runs are reported.

| | RBF | | Matérn 5/2 | | Periodic | | $t$-noise | |
|---|---|---|---|---|---|---|---|---|
| | context | target | context | target | context | target | context | target |
| CNP | $0.972_{\pm0.008}$ | $0.448_{\pm0.006}$ | $0.846_{\pm0.009}$ | $0.206_{\pm0.006}$ | $-0.163_{\pm0.008}$ | $-1.747_{\pm0.023}$ | $0.363_{\pm0.147}$ | $-1.528_{\pm0.068}$ |
| NP | $0.902_{\pm0.009}$ | $0.420_{\pm0.008}$ | $0.774_{\pm0.012}$ | $0.204_{\pm0.010}$ | $-0.181_{\pm0.010}$ | $-1.338_{\pm0.025}$ | $0.442_{\pm0.016}$ | $-0.792_{\pm0.048}$ |
| CNP+DE | 0.995 | 0.521 | 0.878 | **0.313** | **-0.098** | -1.384 | 0.534 | -1.129 |
| BNP | $\mathbf{1.013}_{\pm0.007}$ | $\mathbf{0.526}_{\pm0.005}$ | $\mathbf{0.890}_{\pm0.009}$ | $\mathbf{0.317}_{\pm0.006}$ | $-0.112_{\pm0.007}$ | $\mathbf{-1.082}_{\pm0.011}$ | $\mathbf{0.553}_{\pm0.009}$ | $\mathbf{-0.630}_{\pm0.014}$ |
| CANP | $1.379_{\pm0.000}$ | $0.838_{\pm0.001}$ | $1.376_{\pm0.000}$ | $0.652_{\pm0.001}$ | $0.476_{\pm0.043}$ | $-5.896_{\pm0.134}$ | $1.104_{\pm0.009}$ | $-2.243_{\pm0.031}$ |
| ANP | $1.379_{\pm0.000}$ | $0.842_{\pm0.002}$ | $1.376_{\pm0.000}$ | $0.660_{\pm0.001}$ | $0.600_{\pm0.034}$ | $-4.357_{\pm0.182}$ | $1.125_{\pm0.003}$ | $\mathbf{-1.776}_{\pm0.021}$ |
| CANP+DE | 1.378 | 0.847 | 1.376 | 0.670 | **0.771** | -4.598 | **1.161** | -1.991 |
| BANP | $1.379_{\pm0.000}$ | $\mathbf{0.851}_{\pm0.002}$ | $1.376_{\pm0.000}$ | $\mathbf{0.672}_{\pm0.001}$ | $0.705_{\pm0.016}$ | $\mathbf{-3.275}_{\pm0.114}$ | $1.142_{\pm0.007}$ | $\mathbf{-1.718}_{\pm0.055}$ |

## 4 Related Works

Since the first model CNP [7], there have been several follow-up works to improve NP classes in various aspects. NP [8] suggested to use a global latent variable to model functional uncertainty. ANP [12] further improved the reconstruction quality by employing attention mechanism, and [14] conducted comprehensive comparison and empirically concluded that having global latent variable helps. [21, 24] extended NP to work for sequential data. [16] proposed a consistent NP model mainly using graph neural networks to build conditional probabilities. [9] proposed a translation-equivariant version of NP model using convolution operation in context encoding.

Bootstrap and bagging have been used ubiquitously over many areas in statistical modeling and machine learning. We list a few recent works (especially in the deep learning era) that have benefited from bootstrap and related ideas. Deep ensemble [13] is a special case of bagging (but resampling with replacement) and has been shown to improve predictive accuracy and robustness on various tasks. [20] demonstrated that bootstrapping can improve classification performance on noisy or incomplete labels. [18] showed that bootstrapping can improve exploration in deep reinforcement learning. [17], which proposed the amortized bootstrap, is probably the most similar work to ours. They learn an implicit distribution that generates bootstrap estimates of a parameter of interest, and they show that bagging the bootstrap estimates generated from learned distribution outperforms ordinary bagging. The difference is that the amortized bootstrap targets a single task, meaning that they only learn an implicit bootstrap distribution for a single dataset. On the other hand, BNP meta-learns a network that performs bootstrapping and bagging for any dataset from a particular task distribution.

## 5 Experiments

In this section, we compare the baseline NP classes (CNP, NP, CANP, and ANP) to our models (BNP, BANP) on both synthetic and real-world datasets. We also compare ours against Deep Ensemble (DE) of CNP and CANP [13], in which five identical models are trained with different random initializations and data streams, and averaged for prediction. [2] Following [12], we measured the *context likelihood* $\frac{1}{|c|}\sum_{i\in c}\log p(y_i|x_i, X_c, Y_c)$ measuring the reconstruction quality of the contexts and *target likelihood* $\frac{1}{n-|c|}\sum_{i\notin c}\log p(y_i|x_i, X_c, Y_c)$ measuring the prediction accuracy. NP, ANP, BNP, and BANP were trained with $k = 4$ samples ($z$ for NP and ANP, and bootstrap contexts for BNP and BANP) and tested with $k = 50$ samples. Please refer to Appendix B for further details.

### 5.1 1D Regression

We first conducted 1D regression experiments as in [12]. We trained the models with curves generated from GP with RBF kernels and tested in various settings, including model-data mismatch. More specifically, we tested the models trained with RBF kernel on the data generated from GP with other

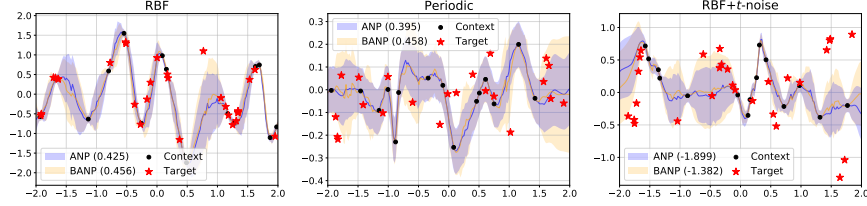

**Figure 2:** Visualization of ANP and BANP for 1D regression data. Ensembled means and ± standard deviations of 50 samples are displayed. The numbers in the legend denotes target log-likelihoods.

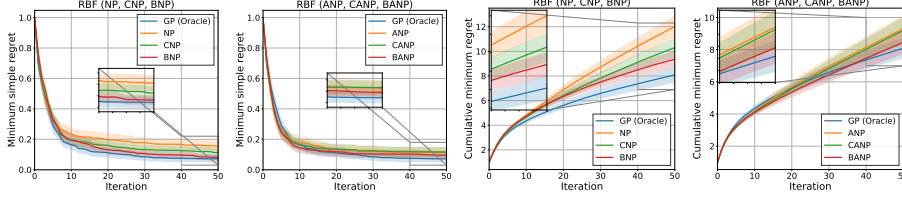

**Figure 3:** Bayesian optimization results for GP prior functions with RBF.

types of kernels (Matérn 5/2, Periodic), and GP with Student's $t$ noise added ($t$-noise). Please refer to Appendix B.1 for a detailed description of network architectures, data generation, training, and testing. For a fair comparison, we set the models to use almost the same number of parameters. Table 1 summarizes the results. BNP and BANP outperformed baselines and even DE, which has $5\times$ the number of parameters. As expected, all models are less accurate in the model-data mismatch setting, but BNP and BANP were affected less, demonstrating the robustness of our approach. Fig. 2 illustrates the behaviour of BANP: ANP and BANP show similar variances for ordinary test data (RBF), but for model-data mismatch data (periodic and $t$-noise), BANP produces wider variances than ANP. We further analyze this aspect by looking at calibrations and sharpness of the predictions in Appendix C.

## 5.2 Bayesian Optimization

We evaluated the models trained in Section 5.1 on Bayesian optimization [2] for functions generated from a GP prior. We reported the best simple regret, which represents the difference between the current best observation and the global optimum, and the cumulative best simple regret for 100 sampled functions. For consistent comparison, we fixed initializations and normalized the results. Results in Fig. 3 show that BNP and BANP consistently achieve lower regret than other NP variants. See Appendix B.2 for more results including model-data mismatch settings.

**Table 2:** EMNIST results. Means and standard deviations of 5 runs are reported.

|  | Seen classes (0-9) | | Unseen classes (10-46) | | $t$-noise | |
|---|---|---|---|---|---|---|
|  | context | target | context | target | context | target |
| CNP | $0.926_{\pm 0.007}$ | $0.751_{\pm 0.005}$ | $0.766_{\pm 0.009}$ | $0.498_{\pm 0.012}$ | $-0.288_{\pm 0.140}$ | $-0.478_{\pm 0.129}$ |
| NP | $0.948_{\pm 0.006}$ | $0.806_{\pm 0.005}$ | $0.808_{\pm 0.005}$ | $0.600_{\pm 0.009}$ | $0.071_{\pm 0.042}$ | $-0.146_{\pm 0.034}$ |
| CNP+DE | 0.954 | 0.813 | 0.818 | 0.616 | **0.107** | **-0.020** |
| BNP | $\mathbf{1.004}_{\pm 0.008}$ | $\mathbf{0.880}_{\pm 0.005}$ | $\mathbf{0.883}_{\pm 0.010}$ | $\mathbf{0.722}_{\pm 0.006}$ | $-0.027_{\pm 0.069}$ | $\mathbf{0.003}_{\pm 0.037}$ |
| CANP | $1.383_{\pm 0.000}$ | $0.950_{\pm 0.004}$ | $1.382_{\pm 0.000}$ | $0.834_{\pm 0.002}$ | $0.133_{\pm 0.196}$ | $-0.492_{\pm 0.108}$ |
| ANP | $1.383_{\pm 0.000}$ | $0.993_{\pm 0.005}$ | $\mathbf{1.383}_{\pm 0.000}$ | $0.894_{\pm 0.004}$ | $0.249_{\pm 0.084}$ | $-0.132_{\pm 0.029}$ |
| CANP+DE | 1.383 | 0.976 | **1.383** | 0.881 | 0.307 | -0.240 |
| BANP | $1.383_{\pm 0.000}$ | $\mathbf{1.010}_{\pm 0.006}$ | $1.382_{\pm 0.000}$ | $\mathbf{0.942}_{\pm 0.005}$ | $\mathbf{0.524}_{\pm 0.102}$ | $\mathbf{0.124}_{\pm 0.060}$ |

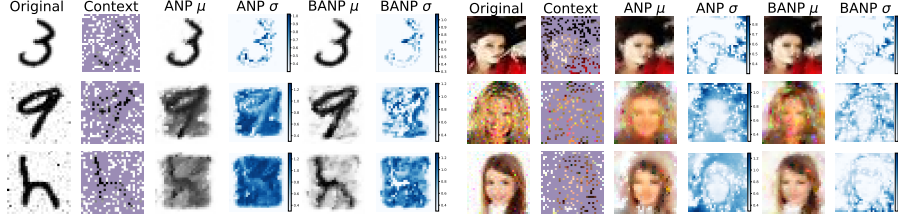

**Figure 4:** ANP vs BANP on EMNIST and CelebA32. The second and third row contains $t$-noises in the image. Ensembled means and standard deviations of 50 samples are displayed.

**Table 3:** CelebA32 results.

|  | Without noise | | $t$-noise | |
|---|---|---|---|---|
|  | context | target | context | target |
| CNP | $2.975_{\pm0.013}$ | $2.199_{\pm0.003}$ | $0.350_{\pm0.384}$ | $-1.468_{\pm0.329}$ |
| NP | $3.066_{\pm0.011}$ | $2.492_{\pm0.014}$ | $0.005_{\pm0.195}$ | $-0.217_{\pm0.104}$ |
| CNP+DE | 3.082 | 2.426 | **1.361** | -0.451 |
| BNP | $\mathbf{3.269}_{\pm0.008}$ | $\mathbf{2.788}_{\pm0.005}$ | $1.224_{\pm0.422}$ | $\mathbf{0.454}_{\pm0.094}$ |
| CANP | $\mathbf{4.150}_{\pm0.000}$ | $2.731_{\pm0.006}$ | $2.985_{\pm0.149}$ | $-0.730_{\pm0.045}$ |
| ANP | $\mathbf{4.150}_{\pm0.000}$ | $2.947_{\pm0.007}$ | $3.037_{\pm0.102}$ | $\mathbf{-0.099}_{\pm0.150}$ |
| CANP+DE | **4.150** | 2.814 | **3.401** | -0.0466 |
| BANP | $4.149_{\pm0.000}$ | $\mathbf{3.129}_{\pm0.005}$ | $3.395_{\pm0.078}$ | $\mathbf{0.083}_{\pm0.126}$ |

**Table 4:** Predator-prey model results.

|  | Simulated | | Real | |
|---|---|---|---|---|
|  | context | target | context | target |
| CNP | $0.088_{\pm0.031}$ | $-0.142_{\pm0.028}$ | $-2.702_{\pm0.007}$ | $-3.013_{\pm0.025}$ |
| NP | $-0.002_{\pm0.039}$ | $-0.252_{\pm0.036}$ | $-2.747_{\pm0.019}$ | $-3.057_{\pm0.020}$ |
| CNP+DE | 0.176 | **-0.026** | -2.670 | **-2.952** |
| BNP | $\mathbf{0.213}_{\pm0.045}$ | $\mathbf{-0.011}_{\pm0.041}$ | $\mathbf{-2.654}_{\pm0.005}$ | $\mathbf{-2.942}_{\pm0.010}$ |
| CANP | $2.573_{\pm0.014}$ | $1.819_{\pm0.021}$ | $1.767_{\pm0.089}$ | $-8.007_{\pm0.538}$ |
| ANP | $2.582_{\pm0.007}$ | $1.828_{\pm0.007}$ | $1.720_{\pm0.257}$ | $-7.809_{\pm0.642}$ |
| CANP+DE | 2.591 | 1.874 | 2.021 | -5.440 |
| BANP | $2.586_{\pm0.009}$ | $1.855_{\pm0.009}$ | $1.783_{\pm0.156}$ | $\mathbf{-5.465}_{\pm0.278}$ |

## 5.3 Image Completion

We compared the models on image completion tasks on EMNIST [3] and CelebA [15] (resized to 32×32). We followed the setting in [8, 12]; see Appendix B.3 for details. As a model-data mismatch setting, we trained the models for EMNIST using the first 10 classes and tested on the remaining 37 classes. We also tested the setting for which Student's $t$-noise were added to the pixel values. We summarize results in Table 2 and Table 3. Except for BNP for EMNIST with $t$-noise setting, ours outperformed the baselines. Fig. 4 compares the completion results of ANP and BANP. ANP often breaks down with noise, while BANP successfully recovers the shapes of objects in images with less blur.

## 5.4 Predator-Prey Model

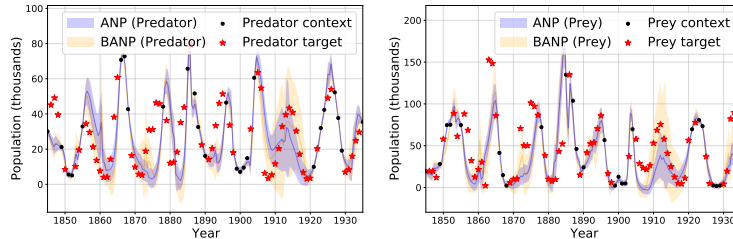

**Figure 5:** ANP vs BANP on Hudson's Bay hare (right)-lynx (left) data. Ensembled means and standard deviations of 50 samples are displayed.

Finally, following [9], we applied the models to predator-prey population data. We first trained the models using simulated data generated from a Lotka-Volterra model [23] and tested on real-world data (Hudson's Bay hare-lynx data). As noted and empirically demonstrated in [9], the hare-lynx data is quite different from the simulated data, so it acts as a mismatch scenario. The results are summarized in Table 4. We obtained the similar results as before; BNP and BANP outperformed the baselines and were comparable to DE for both simulated and real-world data. Fig. 5 shows a similar trend as in Fig. 2; BANP tends to be more conservative for mismatch data by producing larger variances.

# 6  Conclusion

In this paper, we proposed BNP, a novel member of the NP family, which uses bootstrapping to induce functional uncertainty. We demonstrated that BNP could successfully learn robust predictors, especially under model-data mismatch settings. Although not presented here, our model can be applied to any NP variants (or more) seamlessly. For instance, ours can readily be applied to recently proposed convolutional CNP [9]. As future work, one could consider developing a bootstrap resampling algorithm for more general settings. Here we presented an example of using residual bootstrap for regression, but this is not directly applicable for classification. Designing a framework that could "learn" to resample bootstrap datasets in a data-driven way would be an interesting and promising research direction. Finally, we want to stress that the idea of using bootstrap for inducing uncertainty may be useful for many other machine learning problems, especially the ones processing sets of data (e.g., [25]).

## Broader Impact

Uncertainty, robustness, interpretability in predictions have been important desiderata for machine learning algorithms, especially because we have seen actual incidents showing that the algorithms without those could lead to serious damage even threatening human life. The proposed approach suggests a way to enhance robustness by considering uncertainty in data distribution, and the idea of enhancing robustness via bootstrap can be applied to many algorithms over various fields. Therefore, we think that our paper potentially has a positive impact on many areas. Among the experiments we conducted, the predator-prey data experiment (Section 5.4) shows this well, where data generated from a well-established model (Lotka-Volterra model) could be seriously different from real data (Hudson's Bay hare-lynx data), and our model could reduce the risk of failure in such case. However, we admit that the proposed approach may still be vulnerable to various scenario could happen in real life, so should not be treated as an absolute standard to follow. Our model just reduces the probability of failure in a more natural way (i.e., more "data-driven" way).

## Acknowledgments and Disclosure of Funding

This work was supported by Engineering Research Center Program through the National Research Foundation of Korea (NRF) funded by the Korean Government MSIT (NRF-2018R1A5A1059921), Institute of Information & communications Technology Planning & Evaluation (IITP) grant funded by the Korea government (MSIT) (No.2019-0-00075), IITP grant funded by the Korea government(MSIT) (No.2017-0-01779, XAI) and the grant funded by 2019 IT Promotion fund (Development of AI based Precision Medicine Emergency System) of the Korea government (Ministry of Science and ICT). EY is also supported by Samsung Advanced Institute of Technology (SAIT). YWT's research leading to these results has received funding from the European Research Council under the European Union's Seventh Framework Programme (FP7/2007-2013) ERC grant agreement no. 617071.

## Footnotes

[1]Unless specified otherwise, we sample the same number of elements as the original set: $|X| = |\tilde{X}^{(j)}|$.

[2] One could also consider DE of NP or BNP, but here we want to compare the net effect of DE without any other source of uncertainty.

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
