[Supplementary Material]

# Supplementary Material for Bootstrapping Neural Processes

**Juho Lee**[1,2*]**, Yoonho Lee**[2*]**, Jungtaek Kim**[3]**,**
**Eunho Yang**[1,2]**, Sung Ju Hwang**[1,2]**, Yee Whye Teh**[4]
KAIST[1], Daejeon, South Korea, AITRICS[2], Seoul, South Korea,
POSTECH[3], Pohang, South Korea, University of Oxford[4], Oxford, England
juholee@kaist.ac.kr

## A Model Architectures

### A.1 Conditional Neural Process (CNP), Neural Process (NP) and Bootstrapping Neural Process (BNP)

We borrowed most of the architectures from the paper [7] and their source code released [1].

**Encoder** Let $\mathrm{MLP}(\ell, d_{\mathrm{in}}, d_h, d_{\mathrm{out}}), (\ell \geq 2)$ be a multilayer perceptron having the structure

$$
\begin{aligned}
\mathrm{MLP}(\ell, d_{\mathrm{in}}, d_h, d_{\mathrm{out}}) = {}& \mathrm{Linear}(d_h, d_{\mathrm{out}}) \\
& \circ \underbrace{(\mathrm{ReLU} \circ \mathrm{Linear}(d_h, d_h) \circ \dots)}_{\times(\ell-2)} \\
& \circ \mathrm{Linear}(d_h, d_{\mathrm{in}}).
\end{aligned}
\tag{A.1}
$$

An encoder of a NP consists of a deterministic path and a latent path using two identical structures (but with separate parameters),

$$
\begin{aligned}
h_1 &= \frac{1}{|c|} \sum_{i \in c} \mathrm{MLP}(\ell_{\mathrm{pre}}, d_x + d_y, d_h, d_h)([x_i, y_i]), \\
\phi &= \mathrm{MLP}(\ell_{\mathrm{post}}, d_h, d_h)(h_1), \quad f_{\mathrm{denc}}(X_c, Y_c) = \phi \\
h_2 &= \frac{1}{|c|} \sum_{i \in c} \mathrm{MLP}(\ell_{\mathrm{pre}}, d_x + d_y, d_h, d_h)([x_i, y_i]), \\
(\eta, \rho') &= \mathrm{MLP}(\ell_{\mathrm{post}}, d_h, 2d_z)(h_2), \\
\rho' &= 0.1 + 0.9 \cdot \mathrm{sigmoid}(\tilde{\rho}), \quad f_{\mathrm{lenc}}(X_c, Y_c) = (\eta, \rho),
\end{aligned}
\tag{A.2}
$$

where $d_x$ and $d_y$ are the dimensionalities of $x$ and $y$ respectively, and $d_h$ is fixed to 128 for all experiments.

An original CNP uses only one deterministic encoder, but that would perform worse than NP because it uses twice less number of parameters. For a fair comparison, we used two identical encoders for

CNP as well.

$$h_1 = \frac{1}{|c|} \sum_{i \in c} \mathrm{MLP}(\ell_{\mathrm{pre}}, d_x + d_y, d_h, d_h)([x_i, y_i]),$$

$$\phi_1 = \mathrm{MLP}(\ell_{\mathrm{post}}, d_h, d_h)(h_1)$$

$$h_2 = \frac{1}{|c|} \sum_{i \in c} \mathrm{MLP}(\ell_{\mathrm{pre}}, d_x + d_y, d_h, d_h)([x_i, y_i]),$$

$$\phi_2 = \mathrm{MLP}(\ell_{\mathrm{post}}, d_h, d_h)(h_2)$$

$$\phi = [\phi_1, \phi_2], \quad f_{\mathrm{enc}}(X_c, Y_c) = \phi. \tag{A.3}$$

BNP uses exactly the same network encoder as CNP.

**Decoder**  A decoder in CNP and NP take a represerntation of a context and transform it to parameters of conditional probability. Let $x_*$ be a target data point. A decoder of CNP is defined as

$$(\mu, \sigma') = \mathrm{MLP}(\ell_{\mathrm{dec}}, 2d_h + d_x, d_h, 2d_y)([\phi, x_*])$$

$$\sigma = 0.1 + 0.9 \cdot \mathrm{softplus}(\sigma'), \quad f_{\mathrm{dec}}(\phi, x_*) = (\mu, \sigma). \tag{A.4}$$

A decoder for NP uses excatly the same architecture except for that it takes $[\phi, z]$ instead.

$$(\mu, \sigma') = \mathrm{MLP}(\ell_{\mathrm{dec}}, d_h + d_z + d_x, d_h, 2d_y)([\phi, z, x_*])$$

$$\sigma = 0.1 + 0.9 \cdot \mathrm{softplus}(\sigma'), \quad f_{\mathrm{dec}}(\phi, x_*) = (\mu, \sigma). \tag{A.5}$$

BNP uses the same decoder as CNP when computing the deterministic representation without bootstrapping (base model). When decoding an aggregated representations from an original context $\phi$ and a bootstrapped context $\tilde{\phi}$, we add an adaptation layer to the first linear layer of the MLP.

$$h_1 = \mathrm{Linear}(2d_h + d_x, d_h)([\phi, x_*])$$

$$h_2 = \mathrm{Linear}(2d_h, d_h)(\tilde{\phi}) \quad \text{(adaptation layer)}$$

$$(\mu, \sigma') = \mathrm{MLP}(\ell_{\mathrm{dec}} - 1, d_h, d_h, 2d_y)(\mathrm{ReLU}(h_1 + h_2))$$

$$\sigma = 0.1 + 0.9 \cdot \mathrm{softplus}(\sigma'), \quad f_{\mathrm{dec}}(\phi, \tilde{\phi}, x_*) = (\mu, \sigma). \tag{A.6}$$

## A.2  Conditional Attentive Neural Process (CANP), Attentive Neural Process (ANP) and Bootstrapping Attentive Neural Process (BANP)

**Encoder**  An encoder of ANP has a deterministic path and latent path. A deterministic path uses a self-attention and cross-attention to summarize contexts. Let $\mathrm{MHA}(d_{\mathrm{out}})$ be a multi-head attention [12] comptued as follows:

$$Q' = \{\mathrm{Linear}(d_{\mathrm{q}}, d_{\mathrm{out}})(q)\}_{q \in Q}, \quad \{Q'_j\}_{j=1}^{n_{\mathrm{head}}} = \mathrm{split}(Q', n_{\mathrm{head}})$$

$$K' = \{\mathrm{Linear}(d_{\mathrm{k}}, d_{\mathrm{out}})(k)\}_{k \in K}, \quad \{K'_j\}_{j=1}^{n_{\mathrm{head}}} = \mathrm{split}(K', n_{\mathrm{head}})$$

$$V' = \{\mathrm{Linear}(d_{\mathrm{v}}, d_{\mathrm{out}})(v)\}_{v \in V}, \quad \{V'_j\}_{j=1}^{n_{\mathrm{head}}} = \mathrm{split}(V', n_{\mathrm{head}})$$

$$H_j = \mathrm{softmax}(Q'_j(K'_j)^\top / \sqrt{d_{\mathrm{out}}})V'_j, \quad H = \mathrm{concat}(\{H_j\}_{j=1}^{n_{\mathrm{head}}})$$

$$H' = \mathrm{LN}(Q' + H)$$

$$\mathrm{MHA}(d_{\mathrm{out}})(Q, K, V) = \mathrm{LN}(H' + \mathrm{ReLU}(\mathrm{Linear}(d_{\mathrm{out}}, d_{\mathrm{out}}))). \tag{A.7}$$

Here, $(q_{\mathrm{k}}, q_{\mathrm{k}}, q_{\mathrm{v}})$ denotes the dimensionalities of query $Q$, key $K$, and value $V$ respectively, $d_{\mathrm{out}}$ is an output dimension, $n_{\mathrm{head}}$ is a number of heads, split and concat are splitting and concatenating operation in feature axis, and LN is the layer normalization [1]. A self-attention is defined as simply tying $Q = K = V$, $\mathrm{SA}(d_{\mathrm{out}})(X) = \mathrm{MHA}(d_{\mathrm{out}})(X, X, X)$. A deterministic path of ANP is then defined as

$$f_{\mathrm{qk}} = \mathrm{MLP}(\ell_{\mathrm{qk}}, d_x, d_h, d_h)$$

$$q = f_{\mathrm{qk}}(x_*), \quad K = \{f_{\mathrm{qk}}(x_i)\}_{i \in c}$$

$$V = \mathrm{SA}(d_h)(\{\mathrm{MLP}(\ell_{\mathrm{v}}, d_x + d_y, d_h)([x_i, y_i])\}_{i \in c}))$$

$$\phi = \mathrm{MHA}(d_h)(q, K, V), \quad f_{\mathrm{denc}}(X_c, Y_c, x_*) = \phi. \tag{A.8}$$

A latent path of ANP is

$$H = \text{SA}(d_h)(\{\text{ReLU} \circ \text{MLP}(\ell_{\text{pre}}, d_x + d_y, d_h, d_h)([x_i, y_i])\}_{i \in c})$$

$$(\eta, \rho') = \text{MLP}(\ell_{\text{post}}, d_h, 2d_z)\left(\frac{1}{|c|}\sum_{i \in c} h_i\right)$$

$$\rho = 0.1 + 0.9 \cdot \text{sigmoid}(\rho'), \quad (\eta, \rho) = f_{\text{lenc}}(X_c, Y_c). \tag{A.9}$$

For CANP and BANP, we use the same architecture having two paths as follows:

$$f_{\text{qk}} = \text{MLP}(\ell_{\text{qk}}, d_x, d_h, d_h)$$

$$q = f_{\text{qk}}(x_*), \quad K = \{f_{\text{qk}}(x_i)\}_{i \in c}$$

$$V = \text{SA}(d_h)(\{\text{MLP}(\ell_{\text{v}}, d_x + d_y, d_h)([x_i, y_i])\}_{i \in c})$$

$$\phi_1 = \text{MHA}(d_h)(q, K, V)$$

$$H = \text{SA}(d_h)(\{\text{ReLU} \circ \text{MLP}(\ell_{\text{pre}}, d_x + d_y, d_h, d_h)([x_i, y_i])\}_{i \in c})$$

$$\phi_2 = \text{MLP}(\ell_{\text{post}}, d_h, d_h)\left(\frac{1}{|c|}\sum_{i \in c} h_i\right)$$

$$\phi = [\phi_1, \phi_2], \quad f_{\text{enc}}(X_c, Y_c, x_*) = \phi. \tag{A.10}$$

**Decoder**  Decoders are the same is in Appendix A.1.

# B  Experimental Details

## B.1  1D Regression

**Architectures**  For models without attention (CNP, NP, BNP), we set $\ell_{\text{pre}} = 4, \ell_{\text{post}} = 2, \ell_{\text{dec}} = 3, d_h = 128$. For NP we set $d_z = 128$. For models with attention (CANP, ANP, BANP), we set $\ell_{\text{v}} = 2, \ell_{\text{qk}} = 2, \ell_{\text{pre}} = 2, \ell_{\text{post}} = 2, \ell_{\text{dec}} = 3, d_h = 128, n_{\text{head}} = 8$ and $d_z = 128$ for ANP.

**Data generation**  We trained all the models using data generated from Gaussian Processs (GPs) with RBF kernel. For each task $(X, Y, c)$, we first generated $x \overset{\text{i.i.d.}}{\sim} \text{Unif}(-2, 2)$ and generated $Y$ from using RBF Kernel $k(x, x') = s^2 \cdot \exp(-\|x - x'\|^2 / 2\ell^2)$ with $s \sim \text{Unif}(0.1, 1.0)$ and $\ell \sim \text{Unif}(0.1, 0.6)$, and output additive noise $\mathcal{N}(0, 10^{-2})$. The size of the task and the size of the context $c$ was drawn as $|c| \sim \text{Unif}(3, 47)$ and $n - |c| \sim \text{Unif}(3, 50 - |c|)$. For model-data mismatch scenario, we generated data from GP with Matern52 kernels, periodic kernels, and GP with RBF kernel plus Student's $t$ noise. For Matern52 kernel $k(x, x') = s^2(1 + \sqrt{5}d/\ell + 5d^2/(3\ell^2)) \exp(-\sqrt{5}d/\ell), \quad (d = \|x - x'\|)$, we sampled $s \sim \text{Unif}(0.1, 1.0)$ and $\ell \sim \text{Unif}(0.1, 0.6)$. For periodic kernel $k(x, x') = s^2 \exp(-2\sin^2(\pi\|x - x'\|^2/p)/\ell^2)$, we sampled $s \sim \text{Unif}(0.1, 1.0)$ and $\ell \sim \text{Unif}(0.1, 0.6)$ and $p \sim \text{Unif}(0.1, 0.5)$. For Student-$t$ noise, we added $\varepsilon \sim \gamma \cdot \mathcal{T}(2.1)$ to the curves generated from GP with RBF kernel, where $\mathcal{T}(2.1)$ is a Student's $t$ distribution with degree of freedom 2.1 and $\gamma \sim \text{Unif}(0, 0.15)$.

**Training and testing**  We trained all the model for 100,000 steps with each step computes updates with a batch containing 100 tasks. We used Adam optimizer [9] with initial learning rate $5 \cdot 10^{-4}$ and decayed the learning rate using cosine annealing scheme. NP and ANP were trained using $k = 4$ samples for $z$ (as in [2]), and tested with $k = 50$ samples. BNP and BANP were trained with $k = 4$ bootstrap contexts and tested with $k = 50$ samples. The size of the task and the size of the context $c$ was drawn as $|c| \sim \text{Unif}(3, 200)$ and $n - |c| \sim \text{Unif}(3, 200 - |c|)$. Testings were done for 3,000 batches with each batch containing 16 tasks (48,000 tasks in total).

## B.2  Bayesian Optimization

**Architectures / Training and testing**  For these experiments, we followed the settings described in Appendix B.1.

**Prior function generation**    We sampled 100 GP prior functions from zero mean and unit variance. After realizing them, the prior functions are used to optimize via Bayesian optimization. We normalized these functions in order to fairly compare simple regrets and cumulative regrets across distinct sampled functions (Basically, since they are sampled from same distributions, the scales of them are quite similar, but we used more precise evaluations).

**Bayesian optimization setting**    As presented in the Bayesian optimization results, all the methods are started from same initializations. We employed Gaussian process regression [11] with squared exponential kernels as a surrogate model, and expected improvement [6] as an acquisition function, which is optimized by the multi-started local optimization method, L-BFGS-B with 100 initial points. All the experiments are implemented with [8].

## B.3    Image Completion

**EMNIST architectures**    For models without attention (CNP, NP, BNP), we set $\ell_{\text{pre}} = 5, \ell_{\text{post}} = 3, \ell_{\text{dec}} = 4, d_h = 128$. For NP we set $d_z = 128$. For models with attention (CANP, ANP, BANP), we set $\ell_{\text{v}} = 3, \ell_{\text{qk}} = 3, \ell_{\text{pre}} = 3, \ell_{\text{post}} = 3, \ell_{\text{dec}} = 4, d_h = 128, n_{\text{head}} = 8$ and $d_z = 128$ for ANP.

**CelebA32 architectures**    For models without attention (CNP, NP, BNP), we set $\ell_{\text{pre}} = 6, \ell_{\text{post}} = 3, \ell_{\text{dec}} = 5, d_h = 128$. For NP we set $d_z = 128$. For models with attention (CANP, ANP, BANP), we set $\ell_{\text{v}} = 4, \ell_{\text{qk}} = 3, \ell_{\text{pre}} = 4, \ell_{\text{post}} = 3, \ell_{\text{dec}} = 5, d_h = 128, n_{\text{head}} = 8$ and $d_z = 128$ for ANP.

**Data generation**    Each task $(X, Y, c)$ was sampled from an image. Following [3, 7], we sampled 2D coordinates from an image and rescaled the values into $[-1, 1]$ to comprise $X$, and rescaled the corresponding pixel values into $[-0.5, 0, 5]$ to comprise $Y$. The size of the task and the size of the context $c$ was drawn as $|c| \sim \text{Unif}(3, 200)$ and $n - |c| \sim \text{Unif}(3, 200 - |c|)$. For EMNIST we used the first 10 classes during training, and tested on remaining 37 classes as a model-data mismatch scenario.

**Training and testing**    Same as Appendix B.1, except that all the models were trained for 200 epochs through the datasets. The models were tested on entire test set where each sample in a test set comprises a task. For a model-data mismatch scenario with Student's $t$ noise, we added $\varepsilon \sim \gamma \cdot \mathcal{T}(2.1)$ with $\gamma \sim \text{Unif}(0, 0.09)$ to $Y$.

## B.4    Lotka-Volterra

**Architectures**    For models without attention (CNP, NP, BNP), we set $\ell_{\text{pre}} = 4, \ell_{\text{post}} = 2, \ell_{\text{dec}} = 3, d_h = 128$. For NP we set $d_z = 128$. For models with attention (CANP, ANP, BANP), we set $\ell_{\text{pre}} = 2, \ell_{\text{post}} = 2, \ell_{\text{dec}} = 3, d_h = 128, n_{\text{head}} = 8$ and $d_z = 128$ for ANP.

**Dataset generation**    We followed the setting in [4], please refer to the description in the paper. A task $(X, Y, c)$ is then constructed by uniformly subsampling $X$ and corresponding $Y$ from the generated series. The size of the task and the size of the context $c$ was drawn as $|c| \sim \text{Unif}(15, 85)$ and $n - |c| \sim \text{Unif}(15, 100 - |c|)$. Due to the scaling issue, $X$ and $Y$ values were standardized using the statistics computed from the context:

$$x'_i = \frac{x_i - \text{mean}(X_c)}{\text{std}(X_c) + 10^{-5}}, \quad y'_i = \frac{y_i - \text{mean}(Y_c)}{\text{std}(Y_c) + 10^{-5}}. \tag{B.11}$$

**Training and testing**    We trained for 100,000 steps with each step is computed with a batch containing 50 tasks. The other details are the same as in Appendix B.1. Testing was done on 3,000 batches with each batch containing 16 tasks. For real-data testing as a model-data mismatch scenario, following [4], we generated 1,000 batches with each batch containing 16 tasks from Hudson's Bay hare-lynx data. Each task contained $|c| \sim \text{Unif}(15, 76)$ and $n \sim \text{Unif}(15, 91 - |c|)$ points subsampled from the data, and standardized as above.

**Table C.1:** Calibration error and sharpness of the models for 1D regression experiments. Means and standard deviations of 5 runs are reported.

| | RBF | | Matérn 5/2 | | Periodic | | $t$-noise | |
|---|---|---|---|---|---|---|---|---|
| | CE | Sharpness | CE | Sharpness | CE | Sharpness | CE | Sharpness |
| CNP | 0.059±0.003 | 0.072±0.001 | 0.012±0.001 | 0.079±0.001 | 0.171±0.004 | 0.226±0.004 | 0.029±0.002 | 0.093±0.001 |
| NP | 0.016±0.001 | 0.06±0.001 | 0.037±0.005 | 0.067±0.001 | 0.306±0.016 | 0.224±0.001 | 0.138±0.012 | 0.082±0.001 |
| BNP | 0.049±0.002 | 0.069±0.000 | 0.011±0.001 | 0.077±0.000 | 0.145±0.002 | 0.243±0.008 | 0.032±0.001 | 0.098±0.001 |
| CANP | 0.276±0.005 | 0.057±0.001 | 0.127±0.003 | 0.066±0.000 | 0.251±0.022 | 0.157±0.006 | 0.038±0.003 | 0.086±0.002 |
| ANP | 0.144±0.009 | 0.048±0.001 | 0.051±0.003 | 0.055±0.002 | 0.402±0.031 | 0.165±0.007 | 0.154±0.014 | 0.074±0.003 |
| BANP | 0.264±0.001 | 0.057±0.000 | 0.121±0.001 | 0.067±0.000 | 0.0.226±0.002 | 0.176±0.003 | 0.035±0.001 | 0.095±0.001 |

**Table C.2:** Calibration error and sharpness of the models for EMNIST experiments. Means and standard deviations of 5 runs are reported.

| | Seen classes (0-9) | | Unseen classes (10-46) | | $t$-noise | |
|---|---|---|---|---|---|---|
| | CE | sharpness | CE | Sharpness | CE | Sharpness |
| CNP | 0.448±0.007 | 0.035±0.001 | 0.355±0.007 | 0.043±0.001 | 0.066±0.008 | 0.066±0.0.055 |
| NP | 0.423±0.007 | 0.042±0.001 | 0.337±0.004 | 0.050±0.001 | 0.046±0.008 | 0.069±0.001 |
| BNP | 0.435±0.007 | 0.037±0.001 | 0.342±0.006 | 0.046±0.001 | 0.044±0.014 | 0.070±0.003 |
| CANP | 0.533±0.006 | 0.029±0.000 | 0.463±0.003 | 0.032±0.000 | 0.327±0.065 | 0.085±0.006 |
| ANP | 0.489±0.010 | 0.034±0.001 | 0.442±0.008 | 0.036±0.001 | 0.197±0.041 | 0.085±0.006 |
| BANP | 0.511±0.011 | 0.032±0.001 | 0.449±0.006 | 0.035±0.001 | 0.117±0.023 | 0.076±0.006 |

**Table C.3:** Calibration error and sharpness of the models on CelebA32 experiments. Means and standard deviations of 5 runs are reported.

| | Without noise | | $t$-noise | |
|---|---|---|---|---|
| | CE | Sharpness | CE | Sharpness |
| CNP | 0.019±0.000 | 0.056±0.000 | 0.003±0.000 | 0.080±0.002 |
| NP | 0.017±0.000 | 0.065±0.000 | 0.062±0.002 | 0.009±0.003 |
| BNP | 0.008±0.000 | 0.065±0.009 | 0.035±0.006 | 0.101±0.002 |
| CANP | 0.069±0.000 | 0.054±0.000 | 0.007±0.002 | 0.110±0.010 |
| ANP | 0.018±0.000 | 0.062±0.000 | 0.082±0.002 | 0.096±0.001 |
| BANP | 0.018±0.000 | 0.065±0.000 | 0.075±0.012 | 0.100±0.002 |

**Table C.4:** Calibration error and sharpness of the models on Predator-prey experiments. Means and standard deviations of 5 runs are reported.

| | Simulated | | Real | |
|---|---|---|---|---|
| | CE | Sharpness | CE | Sharpness |
| CNP | 0.001±0.000 | 0.578±0.013 | 0.072±0.008 | 1.866±0.058 |
| NP | 0.002±0.003 | 0.567±0.009 | 0.087±0.000 | 1.877±0.069 |
| BNP | 0.003±0.000 | 0.542±0.016 | 0.076±0.011 | 1.975±0.004 |
| CANP | 0.146±0.003 | 0.076±0.001 | 0.565±0.034 | 0.350±0.034 |
| ANP | 0.104±0.004 | 0.064±0.001 | 0.814±0.036 | 0.248±0.015 |
| BANP | 0.140±0.003 | 0.074±0.001 | 0.539±0.039 | 0.352±0.019 |

## C   On calibration and sharpness of the models

We further analyze the learned models using the framework introduced in [10]. Let $\mathcal{T} = (X, Y, c)$ be a task. We see how the predictions for the targets $\{(x_i, y_i)\}_{i \notin c}$ is calibrated, and how large the variances are. Let $F_{x_i}(y_i)$ be the CDF of the prediction $p(y_i | x_i, X_c, Y_c)$. We say a model is perfectly calibrated [10] if for any $p \in [0, 1]$,

$$\frac{1}{n - |c|} \sum_{i \notin c} \mathbb{1}_{\{y_i \leq F_{x_i}^{-1}(p) \leq p\}} \to p \text{ as } n \to \infty, \tag{C.12}$$

The calibration error (CE) is then defined as

$$0 \leq p_1 \leq \dots p_m \leq 1, \quad \hat{p}_\ell = \frac{1}{n - |c|} \sum_{i \notin c}^{n} \mathbb{1}_{\{y_i \leq F_{x_i}^{-1}(p_\ell)\}}, \quad \text{CE}(\mathcal{T}) = \sum_{\ell=1}^{m} (p_\ell - \hat{p}_\ell)^2. \tag{C.13}$$

In our case, we set $p(y_i | x_i, X_c, Y_c) = \mathcal{N}(y_i | \mu_i, \sigma_i^2)$, so

$$F_{x_i}^{-1}(p_\ell) = \mu_i + \sigma_i \sqrt{2} \text{erf}^{-1}(2p_\ell - 1). \tag{C.14}$$

For the models using the ensemble of multiple predictions (NP, ANP, BNP, BANP), we report the ensembled calibration error.

$$(F_{x_i}^{(j)})^{-1}(p_\ell) = \mu_i^{(j)} + \sigma_i^{(j)} \sqrt{2} \text{erf}^{-1}(2p_\ell - 1), \tag{C.15}$$

$$\hat{p}_\ell^{(j)} = \frac{1}{n - |c|} \sum_{i \notin c} \mathbb{1}_{\{y_i \leq (F_{x_i}^{(j)})^{-1}(p_\ell)\}}, \tag{C.16}$$

$$\text{CE}(\mathcal{T}) = \frac{1}{k} \sum_{j=1}^{k} \sum_{i=1}^{n} (p_\ell - \hat{p}_\ell^{(j)})^2. \tag{C.17}$$

We also measure the sharpness [10] which essentially is a average prediction variance.

$$\text{Sharpness}(\mathcal{T}) = \frac{1}{n - |c|} \sum_{i \notin c} \sigma_i^2. \tag{C.18}$$

We evaluated the CE and sharpness of CNP,NP,BNP,CANP,ANP, and BANP trained in the experiments. The results are summarized in Tables C.1 to C.4. In general, ours (BNP and BANP) were better calibrated for model-data mismatch settings, but worse calibrated than NP and ANP for normal test settings or model-data mismatch settings not very different from the normal test setting (e.g., Matérn 5/2 kernels in 1D regression experiments and unseen classes for EMNIST). The reason is that, as we stated in the main text, BNP and BANP tends to produce conservative credible intervals, so become under-confident in normal-test settings and less over-confident in mismatch settings. This corresponds to the observation and theory in [5], where BayesBag is proven to yield credible intervals that are twice larger than the credible intervals produced by normal Bayesian models when the model is correctly specified. The sharpness values also support this claim, where BNP and BANP generally shows higher values than others especially for the mismatch settings. Interestingly, CNP and CANP exhibit similar trends to ours (larger sharpness values than NP or ANP), presumably because they output only one predictor without any functional uncertainty and thus are encouraged to be conservative than NP or ANP to cover wider range predictions. Still, BNP and BANP produced the largest sharpness values in overall. Although this trend we discussed is apparent in 1D regression and predator-prey experiments, we fail to find any of such trend for image completion experiments. We conjecture that this is because for image completion experiments we are restricting the range of function values $y$ to lie in $[-0.5, 0.5]$. This suggests that at least for image completion experiments, the robustness of ours (which is clearly demonstrated both in terms of likelihood values and qualitative samples) comes from a different reason.

## D   Additional results

### D.1   1D Regression

**Ablation study**   We present an ablation study to empirically validate our design choices for BNP and BANP on 1D regression experiment. We compared our full model to the followings: 1) naïve

**Table D.5:** Ablation study for 1D regression.

| | RBF | | Matérn 5/2 | | Periodic | | $t$-noise | |
|---|---|---|---|---|---|---|---|---|
| | context | target | context | target | context | target | context | target |
| BNP | 1.012±0.006 | 0.523±0.004 | 0.891±0.007 | 0.316±0.004 | -0.111±0.002 | -1.089±0.009 | 0.554±0.006 | -0.644±0.010 |
| naïve bootstrap | 0.774±0.015 | 0.304±0.011 | 0.642±0.017 | 0.088±0.008 | -0.261±0.004 | -1.368±0.019 | 0.329±0.012 | -1.203±0.030 |
| - paired bootstrap | 0.990±0.005 | 0.491±0.004 | 0.865±0.006 | 0.269±0.004 | -0.144±0.004 | -1.342±0.014 | 0.455±0.037 | -1.130±0.025 |
| - adaptation layer | 0.900±0.010 | 0.455±0.007 | 0.803±0.011 | 0.294±0.006 | 0.009±0.008 | -0.845±0.006 | 0.579±0.010 | -0.337±0.015 |
| - $p_{\text{base}}$ loss | 0.992±0.010 | 0.496±0.007 | 0.868±0.011 | 0.273±0.007 | -0.135±0.010 | -1.315±0.016 | 0.468±0.014 | -1.068±0.032 |
| BANP | 1.379±0.000 | 0.849±0.001 | 1.376±0.000 | 0.671±0.001 | 0.688±0.044 | -3.429±0.084 | 1.137±0.007 | -1.750±0.031 |
| naïve bootstrap | 1.365±0.008 | 0.822±0.014 | 1.356±0.011 | 0.632±0.014 | 0.502±0.068 | -3.729±0.151 | 1.041±0.023 | -1.782±0.020 |
| - paired bootstrap | 1.379±0.000 | 0.841±0.002 | 1.377±0.000 | 0.655±0.002 | 0.830±0.031 | -4.510±0.138 | 1.141±0.014 | -2.179±0.019 |
| - adaptation layer | 1.370±0.000 | 0.830±0.001 | 1.361±0.000 | 0.639±0.002 | 0.523±0.030 | -3.598±0.099 | 1.046±0.003 | -1.765±0.014 |
| - $p_{\text{base}}$ loss | 1.378±0.000 | 0.836±0.002 | 1.375±0.000 | 0.661±0.001 | 0.647±0.041 | -3.801±0.294 | 1.132±0.004 | -1.697±0.050 |

**Figure D.1:** (Left) processing time per batch. (Right) log-likelihood with different dataset sizes $n$.

residual bootstrap applied to CNP and CANP as described in Section 3.1, 2) BNP and BANP without context resampling via paired bootstrap, and 3) BNP and BANP without adaptation path so decoder just taking the representations of bootstrapped contexts, and 4) BNP and BANP trained without additional $p_{\text{base}}$ loss in Eq. (14). Table D.5 summarizes the results. Except for the case without adaptation layer which showed slightly better performance on mismatch settings, every ablation cases showed poor performance. Naive bootstrap didn't work well for both normal test and mismatch settings, the models without paired bootstrap worked poorly on mismatch settings, and the models without adaptation layer didn't perform well on normal test settings.

**Training time**   We measured averaging training time per batch for CNP, NP, and BNP on 1D regression task (Fig. D.1, left). BNP forwards the data to the model twice, but the actual computation time for BNP is less than the twice of the computation time of NP, because the first pass to compute residuals uses only the context set $(X_c, Y_c)$ which is a subset of the entire batch $(X, Y)$. Thanks to the parallelization by packing every dataset into a tensor, the computation times for all models does not scale linearly with the number of samples $k$.

**Performance for various dataset size $n$**   We measured the average target log-likelihood with varying dataset size $n$ on 1D regression task (Fig. D.1, right). BNP uniformly performed better than CNP and NP by a significant margin.

**Additional figures**   Here we present additional samples in Fig. D.2.

### D.2   Bayesian optimization

Bayesian optimization results, showed in Fig. D.3 demonstrate our methods outperform or are comparable to other methods including GP oracle. For the RBF case, GP oracle is the best result, but our models show the second best results and become comparable to the GP oracle at the last of iterations. On the contrary, in the model-data mismatch setting with $t$-noise (see the second row of

**Figure D.2:** More visualizations for 1D regression experiment.

**Figure D.3:** Bayesian optimization results for GP prior functions with (first row) RBF kernel, (second row) RBF kernel + $t$-noise, (third row) Matérn 5/2 kernel, and (fourth row) Periodic kernel.

Fig. D.3), our methods outperform other methods, which implies that our methods, BNP and BANP are robust to the heavy-tailed noises. Moreover, while CNP and CANP models show the better results in Matérn 5/2 and Periodic cases, our methods are comparable to those methods, as shown in the last two rows of Fig. D.3.

### D.3 Image completion

We present additional visualizations for EMNIST in Fig. D.4 and for CelebA in Fig. D.5.

### D.4 Predator-prey model

We present additional visualizations for predator-prey experiment in Fig. D.6.

## Footnotes

[1] https://github.com/deepmind/neural-processes

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

**Figure D.5:** Image completion results on CelebA32 for ANP and BANP. The results under increasing noise levels are shown.

**Figure D.6:** Regression results for predator-prey data. First two rows shows the results for simulated data, and the last two rows shows the results for the real data (Hudson's Bay hare-lynx data).