[Reviews · NeurIPS 2020]

Review 1

Summary and Contributions: The paper proposes to use bootstrap techniques to improve the uncertainty estimates of Neural Processes (NPs), and consequently, their robustness to model mismatch. The paper states that a naive application of bootstrap to NPs doesn't work, and so it proposes a method that does work.

Strengths: I find the ideas in this paper sound, however, I doubt their significance and relevance. In my understanding, uncertainty estimates of NPs have a very low quality compared to those of GPs. Thus, I would find it significant if a method could close this gap between NPs and GPs. I think that this paper tries to take a step in this direction. However, the bootstrap does not seem to have the power of solving this issue. I think this is best illustrated in Figure 2, where it's clear that there is no big difference between the behaviour of bootstrapped and vanilla models. Therefore, it might have been better to focus on the downstream tasks, where differences could be more visible, for example, as seen in case of Bayesian optimisation in Section 5.2.

Weaknesses: The proposed method, in my opinion, does not provide a good enough solution to the problems related to uncertainty and robusteness in NP-based models. While experiments show improvements in comparison to vanilla NPs, it is unclear to me if these improvements are significant, or whether the evaluation is done correctly. For instance, all tables in the paper report log-likelihoods. However, without extra steps, one can only get the lower bound on the log-likelihood using NPs. I haven't found an explanation on how the numbers in the tables are computed, so I'm still wondering what they are. The figures included in the paper do not show significant improvements over the base models. Thus, if I had to use NPs, I would choose the vanilla version. The extra layer of complexity in the form of bootstrapping, in my opinion, is not worth of the gains. The only result which I find more or less convincing is the Bayesian optimisation experiment. Though, the paper pays very little attention to it, and so it is difficult to interpret the results.

Correctness: I didn't find mistakes in the derivations. Personally, I don't find the method very elegant.

Clarity: yes

Relation to Prior Work: yes

Reproducibility: Yes

Additional Feedback: Some random remarks: -- I find the abstract a bit confusing, e.g. I wouldn't say that NPs learn from a 'data stream' or that having a 'single latent variable limits the flexibility' --line 23: It's weird to see ... in p(y|x....) -- I suspect that the training time of BNPs is higher in comparison to NPs. I think the paper should discuss this. -- I think that if tables like 1 or 2 are used, then they shoud have a lot less entries, e.g. include likelihoods only on the target points. -- the paper lacks the interpretation of Figure 4, which could be quite interesting -------------- POST-REBUTTAL UPDATE ------------------------- I'm grateful to the authors for answering my questions in the rebuttal. I acknowledge that BNPs result in improved log-likelihoods over the baselines. However, I still think that the extra layer of complexity on top of NPs needs to result in even better performance gains before it can be fully justified. From the reviewers' discussion, I believe that other reviewers agree that this is a rather incremental work. I tend to agree with this assessment. This is a borderline paper for me, but I wouldn't be upset if it's accepted.


Review 2

Summary and Contributions: This paper proposes an architecture bootstrapping latent variable on Neural Processes. With a small number of additional parameters, they designed it and validate that it is more robust than Neural Processes, particularly when the test data is from different distribution from a trained one.

Strengths: Motivation is well set, which is addressing the flexibility limitation on the global latent variable and solving it through bootstrap. I think that it deserves to be shared in the community. The proposed method is clearly described, and the experiment design and analysis are also quite good.

Weaknesses: The limitation of the method is that It requires more computation resource even though it just requires a few additional parameters because it processes encoding-decoding twice with k variables. The limitation on the paper is lack of analysis when not working well.

Correctness: Correct

Clarity: Well written

Relation to Prior Work: Yes

Reproducibility: Yes

Additional Feedback: - They mentioned naiive application of bootstrap to NP, but not showed the empirical results about that. As one of baselines, if they compare the result with BNP, it would be better. - BNP is processed in a parallel manner and requires the encoding-decoding twice, which requires more computation time, so analysis about wall time comparison on training can be fruitful to understand BNP. - line 92, in equation, \tilde{X}^{(k)} -> \tilde{X}^{(j)} # To authors: Thank you for updating about processing time. What I wanted to check was convergence speed comparison with baselines, but your data is also wealthy to check.


Review 3

Summary and Contributions: This paper proposes an extension to NPs that is one way to model functional uncertainty of NP models. The proposed method is using bootstrap, which essentially doing re-sampling the contexts and construct an ensemle of predictions. The resulting algorithm is called Bootstrapping Neural Process (BNP). This functional uncertainty modeling is said to improve the robustness of NPs, especially in the case of model-data mismatch.

Strengths: The paper pursues an interesting research question that looks at the problem of model-data mismatch. Modeling functional uncertainty of context representation would be a good way to improve the prediction of target data. The author proposes a method using bootstraping to obtain an ensemle of context representation, hence receives an ansembled distribution of target predictions. The proposed idea makes sense as boostraping has been used successfully in frequentist statistics and other ML frameworks.

Weaknesses: Given the paper's current state, I have following major comments: - The proposed method's motivation is to tackle the issue of model-data mismatch by modeling the context representation uncertainty. However the notion of the model-data mismatch is loosely defined. It would be more interesting if the paper's formulation would fomulate this problem in a principled way, e.g. the model-data mismatch problem can be framed in a more principled way, e.g. training task distribution and target task distribution could be defined on different domains as shown in experiments. - The choice of the training objective in (14) needs justifications. The combined objective of two models with/without bootstraps is somewhat questionable. The computation of residuals would influence a lot to the input hence the convergence of the full model. I would really like to see how this technical issue can be analysed and evaluated with ablation more carefully. - The proposed method seem to enjoy many advantages as seen in 3.4. discussion. But the missing of a parallel implementation and its benefit demonstration would be unfortunate. - BNP/BANP does not always perform better then the original NP family. Sometimes it performes better, but by not much and not clearly seen especially in qualitative results like in EMNIST. - As BNP can model uncertainty of context data, it might be interesting to see comparisons among methods on a different amount of context points.

Correctness: The claim and empirical methodology are correct.

Clarity: The paper is well written, which is easy to follow.

Relation to Prior Work: Yes, the paper has a great discussion to related work.

Reproducibility: Yes

Additional Feedback: ----------------------------------------------------------------------------------- I have read the author response. I have changed my score upon some good responses.


Review 4

Summary and Contributions: The paper proposes a solution to the problem of generating functional uncertainty in neural process models. To achieve this, it uses residual bootstrap. A positive effect on the resulting models is their robustness against model-data mismatch.

Strengths: This work is well-designed and not trivial, as illustrated by the fact that the naïve application of residual bootstrap to NP does not work, sec 3.1. The experimental validation is good, covering several very different scenarios and demonstrating consistently good results. The method rests on a few good ideas, and so should not pose specific reproducibility problems.

Weaknesses: Theoretical analysis and analytical experiments could be strengthened.

Correctness: Yes. The experimental part is good. The experiments are not novel, in the sense that they follow established precedents, but the coverage is good. The code accompanying the paper is complete.

Clarity: The paper is well written and clear overall. Several English errors should not have made it to the submitted paper, but do not obfuscate the meaning of the text. The structure of the paper is good. Though not part of the assessment done for this review, and not necessary, the supplementary material is interesting.

Relation to Prior Work: Yes. The connection is pretty straightforward with the NP literature. There could be a few more connections with the literature built regarding neural architecture aspects. The connection with the literature using bootstrap ideas is fine.

Reproducibility: Yes

Additional Feedback: The paper should be spell-checked and grammar-checked. Line 105 and 108, I believe the ^ is missing from mentions of mu and sigma. Last author of ref 4 missing. ---- note after reviewer discussion I have carefully read the author feedback, other reviews, and the reviewer discussion. My overall score is updated to a solid 7, with higher confidence than before.

[Author Response · NeurIPS 2020]

We thank all the reviewers for their constructive comments. Although the reviewers were generally happy with the
novelty of our method, there were some concerns on the experimental results for which we address in this rebuttal.

**[All reviewers] Typos**  We appreciate for
pointing them out, and will correct them in
the revised version.

**[R1] Improvements over vanila NP**  We
respectfully disagree. Please note that BNP
and BANP outperform baselines in most ex-
periments in terms of log-likelihood, espe-
cially for mismatch data. Even though the
absolute difference may look small in our
metric, the differences are significant because

(a) $k$ vs trainng time

| | context | target |
|---|---|---|
| BNP | $1.012_{\pm 0.006}$ | $0.523_{\pm 0.004}$ |
| $-p_{\text{base}}$ | $0.981_{\pm 0.008}$ | $0.487_{\pm 0.007}$ |
| BANP | $1.379_{\pm 0.000}$ | $0.849_{\pm 0.001}$ |
| $-p_{\text{base}}$ | $1.378_{\pm 0.000}$ | $0.836_{\pm 0.003}$ |

(b) Loss ablation

(c) Context $n$ vs LL

they are 1) logs of likelihoods and 2) measured per datapoint. Note that confidence intervals are on the order of $10^{-3}$
and that our model matches or even outperforms the ensemble of five models. Fig 2. shows the difference in uncertainty
quantification of ANP and BANP. The two models show similar behavior for normal RBF data, but BANP produces
wider credible intervals for mismatch data (Periodic and $t$-noise). In other words, BANP tends to be less confident
for mismatch data and thus better calibrated. We further demonstrate this tendency in the additional qualitative results
given in Fig D. 5. in the supplementary.

**[R1] Why would one consider using BNP despite additional computation?**  We stress again that BNP/BANP show
clear improvements over baseline in terms of log-likelihoods. Another benefit is generality: one can apply the bootstrap
idea to any NP model (e.g., convolutional CNP) without having to carefully design variational distributions and tune the
hyperparameters to train them properly (e.g., choosing latent dimensions, KL annealing, ...).

**[R1] Is log-likelihood a proper performance measure? How they are computed?**  Log-likelihood is a proper
scoring rule (ref [13]) that gives a higher value for a better calibrated model. We computed log-likelihood values
following the convention in the NP literature (ref [14]). As you pointed out, the log-likelihood values computed are
lower-bounds, but they approach the true values as the number of samples $k$ increases. The log-likelihood values
reported in the paper were computed with $k = 50$ samples. We will make this more clear in our revised version.

**[R1] Bayesian optimization is less highlighted**  The bayesian optimization experiment quantifies the quality of the
uncertainty estimates of models through the minimum simple regret and cumulative minimum regret metrics. In
this experiment, BNP/BANP ourperformed other NP baselines and was even comparable to GP. We will discuss and
highlight this more in our revised version.

**[R1, R2] Training time**  We measured the average processing time per batch for CNP, NP, and BNP in (1a). The
computation time of BNP is less than twice of CNP and NP because the first pass to compute residuals uses only the
context set $(X_c, Y_c)$, which is a subset of the entire batch. Thanks to the parallelization, the computation time for NP
and BNP scales sub-linearly with $k$. We will discuss computing time requirements more in the revised version.

**[R2] Comparison to naïve bootstrap**  Please refer to Table D.5 in the supplementary, where we performed ablation
studies including the naïve bootstrap.

**[R2] Failure cases**  We agree that the analysis of failure cases can improve the understanding of our model. Though
not exactly a failure, we think that our models for image completion tasks have room for improvement since we are
restricting the output range to be in $[-1, 1]$, but we do not consider this during residual resampling.

**[R4] Definition of model-data mismatch**  Thanks for pointing this out. For now, we roughly define model-data
mismatch to be the case where the test task distribution differs from the training task distribution. As you mentioned,
the difference can be in terms of domains, or even in generative processes within the same domain. We will add more
discussion on this matter in the revised version.

**[R4] Justification for the objective (14)**  We empirically confirmed that the objective without $p_{\text{base}}$ performs bad
(still better than CNP or NP). Partial results for 1D regression are in (1b); we will include full results in the revised
version.

**[R4] BNP/BANP do not always performs better**  Yes, but please note that ours perform better than baselines for
the most of the cases, especially for mismatch settings. The qualitative results on EMNIST are well reflected in the
log-likelihood values, showing significant improvements over baselines.

**[R4] Benefit from parallelization?**  Our implementation is already utilizing parallelization by packing multiple
bootstrap contexts into a single tensor and process them in parallel. A naïve iterative implementation scales poorly and
takes horribly long to train. Please refer to our source code for implementation details.

**[R4] Number of context points vs performance**  Thanks for the suggestion. In (1c), we measured the target log-
likelihood of CNP, NP, and BNP on 1D regression tasks with varying task size $n$ (thus varying number of contexts).
BNP consistently performed better with a significant margin. Although we did not put it here due to space constraints,
the same was true of models with attention.

**[R5] Proofread**  Sorry for the inconvenience, we will do our best to revise our paper.

[Meta-Review · NeurIPS 2020]

This is an important paper on uncertainty quantification. However as the reviewers noted the main concerns are competitiveness with reespect to GPs and also an analysis (perrhaps with intuitions) of when the method underperforms would be useful. Overall, this paper might pave the way for really interesting follow-ups which will build on top of it.